# Global epidemiology of occult hepatitis B virus infections in blood donors, a systematic review and meta-analysis

Guy Roussel Takuissu[1], Sebastien Kenmoe[2]*, Marie Amougou Atsama[3], Etienne Atenguena Okobalemba[4], Donatien Serge Mbaga[5], Jean Thierry Ebogo-Belobo[6], Arnol Bowo-Ngandji[5], Martin Gael Oyono[7], Jeannette Nina Magoudjou-Pekam[8], Ginette Irma Kame-Ngasse[6], Elisabeth Zeuko'o Menkem[9], Abdel Aziz Selly Ngaloumo[8], Agnès Thierry Rebecca Banlock[5], Alfloditte Flore Feudjio[8], Cromwel Zemnou-Tepap[8], Dowbiss Meta-Djomsi[3], Gilberte Louise Nyimbe Mviena[5], Ines Nyebe Eloundou[5], Jacqueline Félicité Yéngué[10], Josiane Kenfack-Zanguim[8], Juliette Laure Ndzie Ondigui[5], Ridole Martin Zekeng Mekontchou[11], Sabine Aimee Touangnou-Chamda[5], Yrene Kamtchueng Takeu[6], Jean Bosco Taya-Fokou[5], Chris Andre Mbongue Mikangue[5], Raoul Kenfack-Momo[8], Cyprien Kengne-Nde[12], Seraphine Nkie Esemu[2], Richard Njouom[11], Lucy Ndip[2]

1 Centre for Food, Food Security and Nutrition Research, Institute of Medical Research and Medicinal Plants Studies, Yaounde, Cameroon, 2 Department of Microbiology and Parasitology, University of Buea, Buea, Cameroon, 3 Centre de Recherche sur les Maladies Émergentes et Re-Emergentes, Institut de Recherches Médicales et d'Etudes des Plantes Médicinales, Yaounde, Cameroon, 4 Faculty of Medicine and Biomedical Science, The University of Yaounde I, Yaounde, Cameroon, 5 Department of Microbiology, The University of Yaounde I, Yaounde, Cameroon, 6 Medical Research Centre, Institute of Medical Research and Medicinal Plants Studies, Yaounde, Cameroon, 7 Centre for Research on Health and Priority Pathologies, Institute of Medical Research and Medicinal Plants Studies, Yaounde, Cameroon, 8 Department of Biochemistry, The University of Yaounde I, Yaounde, Cameroon, 9 Department of Biomedical Sciences, University of Buea, Buea, Cameroon, 10 Department of Animals Biology and Physiology, The University of Yaounde I, Yaounde, Cameroon, 11 Virology Department, Centre Pasteur of Cameroon, Yaounde, Cameroon, 12 Epidemiological Surveillance, Evaluation and Research Unit, National AIDS Control Committee, Douala, Cameroon

* sebastien.kenmoe@ubuea.cm

**Data Availability Statement:** All relevant data are within the paper and its Supporting information files.

## Abstract

This study aimed to assess the global prevalence of occult hepatitis B in blood donors. We searched PubMed, Web of Science, Global Index Medicus, and Excerpta Medica Database. Study selection and data extraction were performed by at least two independent investigators. Heterogeneity ($I^2$) was assessed using the $\chi^2$ test on the Cochran Q statistic and H parameters. Sources of heterogeneity were explored by subgroup analyses. This study is registered with PROSPERO, number CRD42021252787. We included 82 studies in this meta-analysis. The overall prevalence of OBI was 6.2% (95% CI: 5.4–7.1) in HBsAg negative and anti-HBc positive blood donors. Only sporadic cases of OBI were reported in HBsAg negative and anti-HBc negative blood donors. The overall prevalence of OBI was 0.2% (95% CI: 0.1–0.4) in HBsAg negative blood donors. The prevalence of OBI was generally higher in countries with low-income economic status. The results of this study show that despite routine screening of blood donors for hepatitis B, the transmission of HBV by blood remains possible via OBI and/or a seronegative window period; hence there is a need for

**Funding:** This project is part of the EDCTP2 programme supported by the European Union under grant agreement TMA2019PF-2705. The funders had no role in study design, data collection and analysis, decision to publish, or preparation of the manuscript.

**Competing interests:** The authors have declared that no competing interests exist.

**Abbreviations:** HBsAg, Hepatitis B surface antigen; HBV, Hepatitis B virus; OBI, Occult hepatitis B infection.

active surveillance and foremost easier access to molecular tests for the screening of blood donors before transfusion.

## 1. Introduction

Hepatitis B virus (HBV) infections show a negative window (HBsAg and anti-HBc negative) at the onset of infection. There are also so-called occult HBV infections (OBI) where HBsAg is negative while DNA is positive in serum or liver [1–5]. These two factors contribute to making HBV the most frequent virus whose transmission during blood transfusion is high [6]. After the serologically negative period, individuals infected with HBV will develop HBsAg normally while subjects with OBI remain negative. Occult HBV infections may contribute to the progression of liver disease leading to the development of hepatocellular carcinoma [7–9]. Occult HBV infections are described by several mechanisms among which reduction, modification of the configuration or lack of synthesis of HBsAg due to the mutation of the S gene, formation of HBsAg/anti-HBs complexes, archiving of HBsAg in peripheral blood mononuclear cells, and co-infection with HCV or HIV which could interfere with the HBV replication [10–14]. The risk of transmission of HBV through blood transfusion depends on the HBV endemicity, regions and pre-transfusion tests [6]. In high-income countries, molecular (DNA) and serological (HBsAg, anti-HBc) assays are adopted for screening blood before transfusion [15–18]. The algorithms with minipool testing of 6 to 24 blood donors were introduced due to the low concentration of HBV DNA in OBI cases. In most low-income countries, screening before transfusion remains essentially serological (HBsAg alone or with anti-HBc/anti-HBs) [19] and mostly associated with insufficient blood supply. Some algorithms have considered transfusion of blood with low anti-HBc level [20]. Although OBI is not always transmissible through blood transfusion, several studies have reported cases of post-transfusion infections during OBI or the seronegative window period [5, 21–23]. This residual risk of post-transfusion transmission is greater for OBI donors with a high viral load and those without anti-HBs. The risk of transmission is also important in immunocompromised recipients and multiple transfused patients [1]. Many studies around the world have reported the prevalence of OBI in blood donors. This prevalence greatly varies among these studies, however, no study has yet reported this global prevalence. This systematic review aimed to determine the prevalence of OBI in blood donors worldwide.

## 2. Materials and methods

### 2.1. Protocol and registration

This study was conducted according to the PRISMA guidelines (S1 Appendix) [24]. The protocol was registered in the International Prospective Register of Systematic Reviews (PROSPERO, no. CRD42021252787).

### 2.2. Eligibility criteria

This review considered all observational studies (cross-sectional, case-control, and cohort) written in English or French and limited to blood donors. According to the data provided by the authors of the included articles, the blood donors were classified as 1) OBI in HBsAg negative and anti-HBc positive blood donors, 2) OBI in HBsAg negative and Anti-HBc negative blood donors, and 3) OBI in HBs negative blood donors. All studies regardless of the technique used to detect DNA in blood donors were included. Studies with repeated donors and pooled

test samples were excluded [25]. Studies for which the abstract or full text was not available, duplicates, comments, case reports, case series, and studies fewer than 10 participants, were excluded.

## 2.3. Data sources and search strategy

PubMed, Excerpta Medica Database (Embase), Web of Science, and Global Index Medicus were searched to identify relevant studies published worldwide from database inception to July 2021. The search terms were related to OBI and Blood Donors. The search strategy conducted in PubMed is presented in S2 Appendix. A manual search was also conducted to browse reference lists of eligible articles and other relevant review articles.

## 2.4. Study selection

Duplicates identified in the full list of studies were removed. The articles titles and abstracts retrieved from the electronic and manual literature searches were independently reviewed by the authors for eligibility. Articles deemed potentially eligible, full-text retrieval was performed. The authors independently assessed the completeness of each of these texts for final inclusion. Discussion and consensus were reached to resolve any disagreements observed during study selection.

## 2.5. Data extraction and management

Data from the included studies were extracted using a pre-designed Google data abstraction form. For each eligible article, the extraction was performed independently by at least 2 investigators and verified by a third if necessary. The extracted data were: the name of the first author, year of publication, study design, sampling method, time of data collection and analysis, country, country income level, sample collection period, study site, category of blood donors (HBsAg negative and anti-HBc positive, HBsAg negative and anti-HBc negative or HBsAg negative), age range, methods of diagnosing HBV, laboratory samples used, sample size, number of blood donors with HBV, and risk of bias. In studies where multiple tests for HBsAg were performed, the results of the first test performed was chosen. In studies where multiple molecular detection tests were performed on the same participants, the result with the best sensitivity were selected. Disagreements among the review authors were reconciled by consensus.

## 2.6. Quality assessment

The methodological quality of the included studies was assessed using the tool developed by Hoy et al. for prevalence studies (S3 Appendix) [26].

## 2.7. Statistical analysis

As a single study could report the OBI prevalence for several population categories, the analyses according to the number of prevalence data was conducted. Study-specific estimates were pooled using a random-effect model meta-analysis from DerSimonian and Laird. Heterogeneity was assessed by the Cochrane Q statistical test and quantified by $I^2$ values, assuming that the $I^2$ values of 25%, 50% and 75% represent low, moderate, and high heterogeneity, respectively [27]. Publication bias was assessed by Egger's test and the funnel plot [28]. Subgroup analyses were conducted according to study design, sampling approach, country, WHO and UNSD (United Nations Statistics Division) regions, country income level, OBI categories, and OBI detection assay. A sensitivity analysis including only studies with a low risk of bias and

cross-sectional studies representing the best design for prevalence studies was conducted. A p value <0.05 indicated a significant difference. R software version 4.1.0 was used to perform the analyses [29, 30].

## 3. Results

### 3.1. Study selection

The database search yielded 1011 articles (Fig 1). Subsequently, 241 duplicates were eliminated and from the remaining 770, 579 were excluded due to irrelevant titles and abstracts. A total of 191 eligible articles were therefore fully reviewed with109 excluded with individual reasons given in S4 Appendix. A total of 82 articles were included in the study meeting the defined eligibility criteria, corresponding to 87 blood donor prevalence data (S5 Appendix).

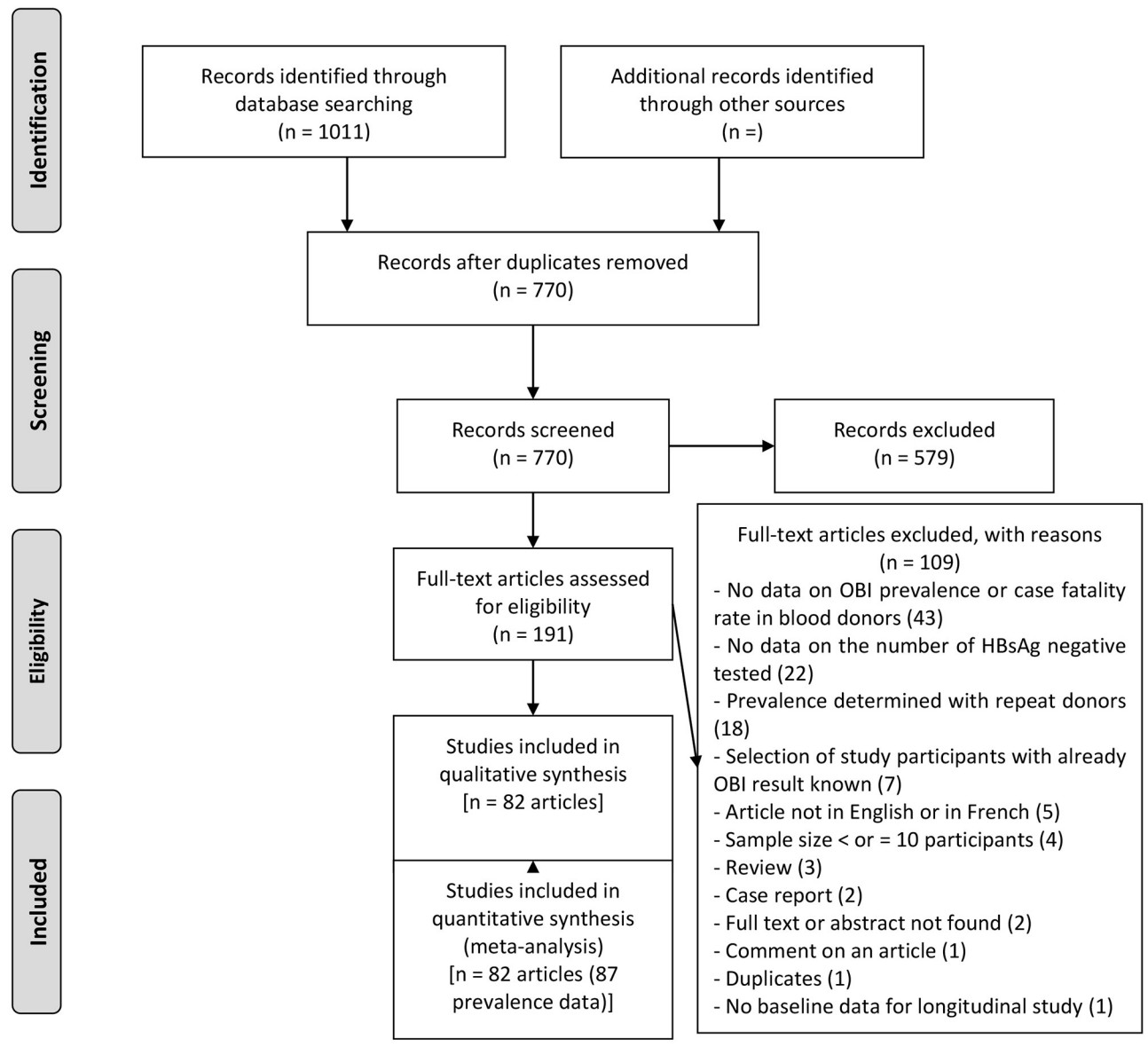

**Fig 1. PRISMA flow-chart of studies selected for the meta-analysis.**

### 3.2. Study characteristics

The aggregated and individual characteristics of the included studies are reported in S6 and S7 Appendices. In most of the studies, the age range of the participants was not reported or not clearly defined (n = 64 or 73.6%), but mainly comprised of adults for the studies that reported the information. The most represented UNSD region was South Asia (n = 27 or 31.0%), closely followed by East Asia (n = 19 or 21.8%); while the most represented WHO regions were the West Pacific (n = 23 or 26.4%) and East Mediterranean (n = 22 or 25.3%). In terms of countries, China was the most represented (n = 15 or 17.2%), followed by India (n = 13 or 14.9%), and Iran (n = 10 or 11.5%). Upper middle income countries were the most represented in the various data (n = 43 or 49.4%). Most of the prevalence data came from cross-sectional studies (n = 86 or 98.9%). Most of the data came from non-probability sampling (n = 71 or 81.6%), with prospective data collection and analysis (n = 67 or 77.0%). The bulk of the data were from a single site (n = 67, 77.0%) and were obtained at the hospital (n = 86, 98.9%). The inclusion period for participants was 1991 to 2019. All data were obtained from a blood sample (n = 87, 100%). The most commonly used diagnostic method was real-time PCR (n = 59, 67.8%). Of the 87-prevalence data studied, more were at low risk of bias (n = 49 or 56.3%) (S8 Appendix).

### 3.3. Occult hepatitis B virus prevalence in HBsAg negative and Anti-HBc positive blood donors

The prevalence of OBI in HBsAg negative and Anti-HBc positive blood donors was assessed in 54 prevalence data from 27 countries (Fig 2 and S9 Appendix). The overall prevalence of OBI in HBsAg negative and Anti-HBc positive blood donors was 6.2% (95% CI: 5.4–7.1). The prevalence of OBI ranged from 0.7% (Europe) to 16.7% (South-east Asia). Occult hepatitis B virus prevalence in HBsAg negative and Anti-HBc positive blood donors was 16.7% (95% CI: 6.0–31.2) in South-east Asia, 10.6% (95% CI: 5.3–17.4) in the Western Pacific, 10.0% (95% CI: 5.0–16.4) in the Eastern Mediterranean, 2.9% (95% CI: 1.6–4.6) in America, 2.8% (95% CI: 0.1–8.3) in Africa, and 0.7% (95% CI: 0.0–6.8) in Europe.

### 3.4. Occult hepatitis B virus prevalence in HBsAg negative and Anti-HBc negative blood donors

The prevalence of OBI in HBsAg negative and Anti-HBc negative blood donors was assessed in 6 prevalence data from 6 countries (Fig 3). Only two studies in Argentina (3/70063) and Egypt (2/238) reported positive OBI cases in HBsAg negative and anti-Hbc negative blood donors.

### 3.5. Occult hepatitis B virus prevalence in HBsAg negative blood donors

The prevalence of OBI in HBsAg negative blood donors was assessed in 27 prevalence data from 11 countries (Fig 4 and S10 Appendix). The overall prevalence of OBI in HBsAg negative blood donors was 0.2% (95% CI: 0.1–0.4). Occult hepatitis B virus prevalence in HBsAg negative blood donors was 5.0% (95% CI: 0.7–12.6) in Africa, 1.2% (95% CI: 0.0–8.3) in the Eastern Mediterranean, 0.0% (95% CI: 0.0–0.1) in the Western Pacific, 0.0% (95% CI: 0.0–0.1) in South-East Asia, and 0.0% (95% CI: 0.0–3.8) in the Americas.

### 3.6. Heterogeneity and publication bias

The degree of heterogeneity and publication bias within the prevalence data is presented in Table 1 and S11–S13 Appendices. In HBsAg negative and Anti-HBc positive blood donors, the estimation of prevalence data was associated with significant heterogeneity and the presence of

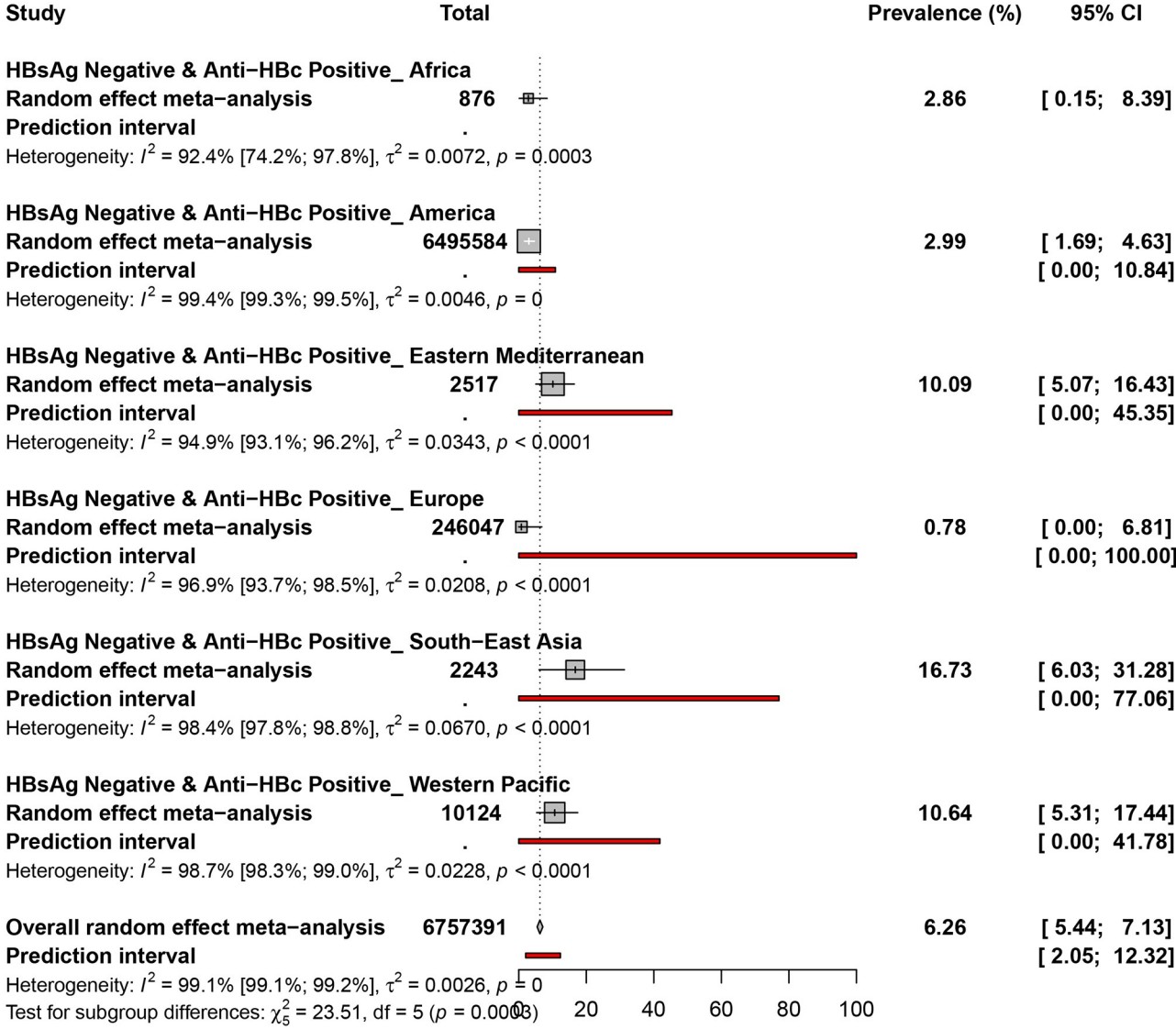

**Fig 2. The pooled global prevalence of occult hepatitis B infection in HBsAg negative and anti-HBc positive blood donors.**

publication bias. The publication bias results obtained by Egger's test were confirmed by the funnel plot (S11 Appendix). In HBsAg negative and Anti-HBc negative blood donors, the estimation of prevalence data was associated with no significant heterogeneity and publication bias. The publication bias results obtained by Egger's test were confirmed by the funnel plot (S12 Appendix). In HBsAg negative blood donors, the estimation of prevalence data was associated with significant heterogeneity (H>1 and I2 >50%) and the presence of publication bias (P < 0.05 for Egger's test). The publication bias results obtained by Egger's test were confirmed by the funnel plot (S13 Appendix).

## 3.7. Subgroup analyses

Among OBI in blood donors, subgroup analysis showed that the overall prevalence was significantly different according to sampling approach (p = 0.003, higher prevalence in probabilistic sampling), countries (p < 0.001, higher prevalence in Malaysia), WHO region (p < 0.001,

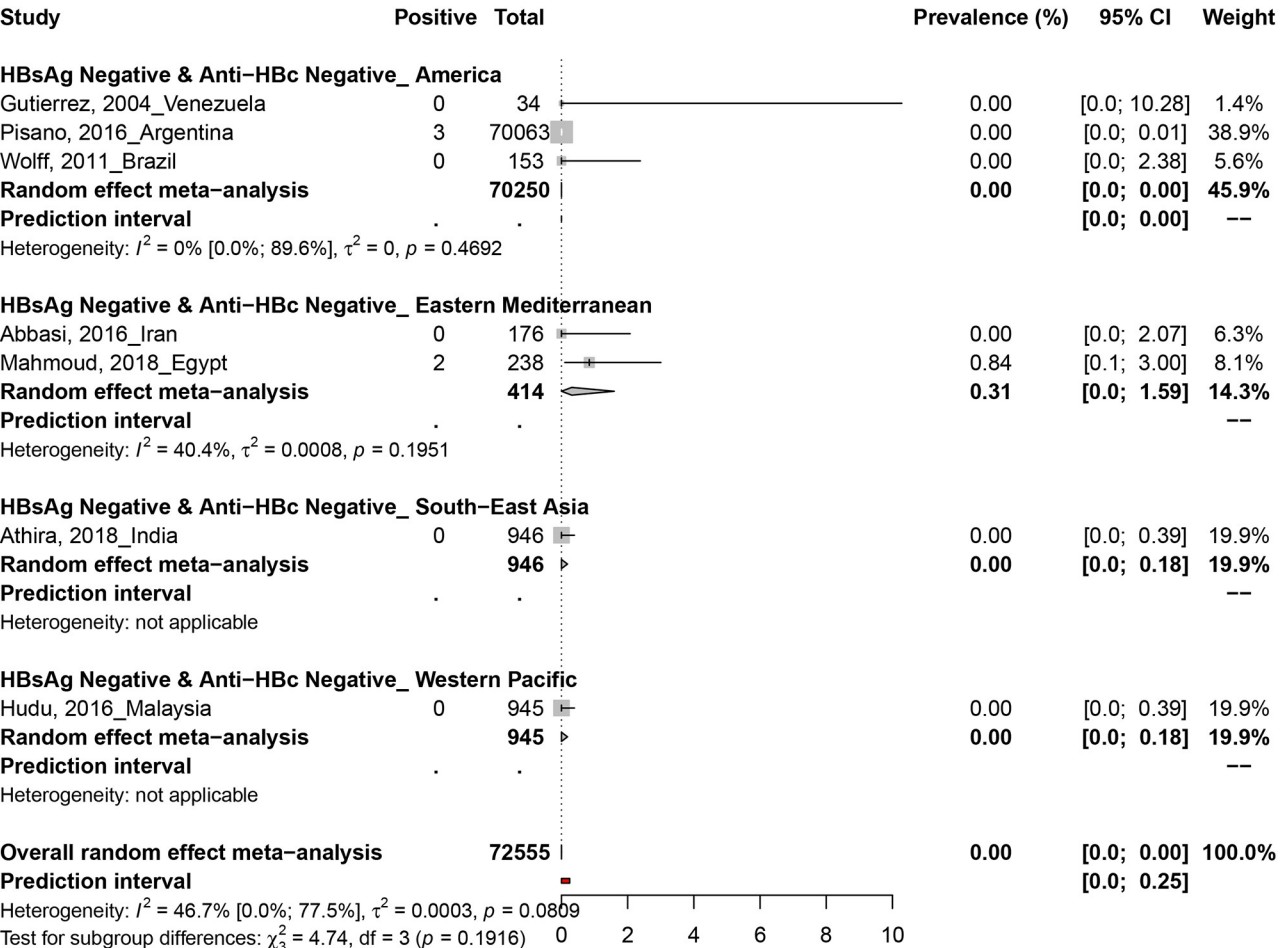

**Fig 3. The pooled global prevalence of occult hepatitis B infection in HBsAg negative and anti-HBc negative blood donors.**

higher prevalence in Eastern Mediterranean, South-East Asia, and Africa), UNSD region (p < 0.001, higher prevalence in Southeastern Asia and Northern Africa), country income level (p < 0.001, higher prevalence in Low-income economies followed by Lower-middle income economies, Upper-middle-income economies, and High-income economies), OBI categories (p <0.001, higher prevalence in HBsAg negative and Anti-HBc positive blood donors), and OBI diagnostic method (p < 0.001, higher prevalence with loop mediated isothermal amplification) (Fig 5, S13 Appendix).

## 4. Discussion

To date, the transmission of HBV through blood donation remains crucial. The pretransfusion test which relies on the detection of HBsAg in serum commonly used mainly in low-resource countries does not rule out subjects in the seronegative window and those with OBI. This study aimed to assess the prevalence of OBI in blood donors. The overall prevalence of OBI in HBsAg negative and Anti-HBc positive blood donors was 6.2%. Only 5 sporadic cases of OBI in HBsAg negative and anti-HBc negative blood donors was obtained. The overall prevalence of OBI in HBsAg negative blood donors was 0.2%. The highest prevalence of OBI was recorded in low-resource countries and in the WHO regions of Eastern Mediterranean, South-east Asia, and Africa. Only two systematic reviews previously reported data on the national prevalence of

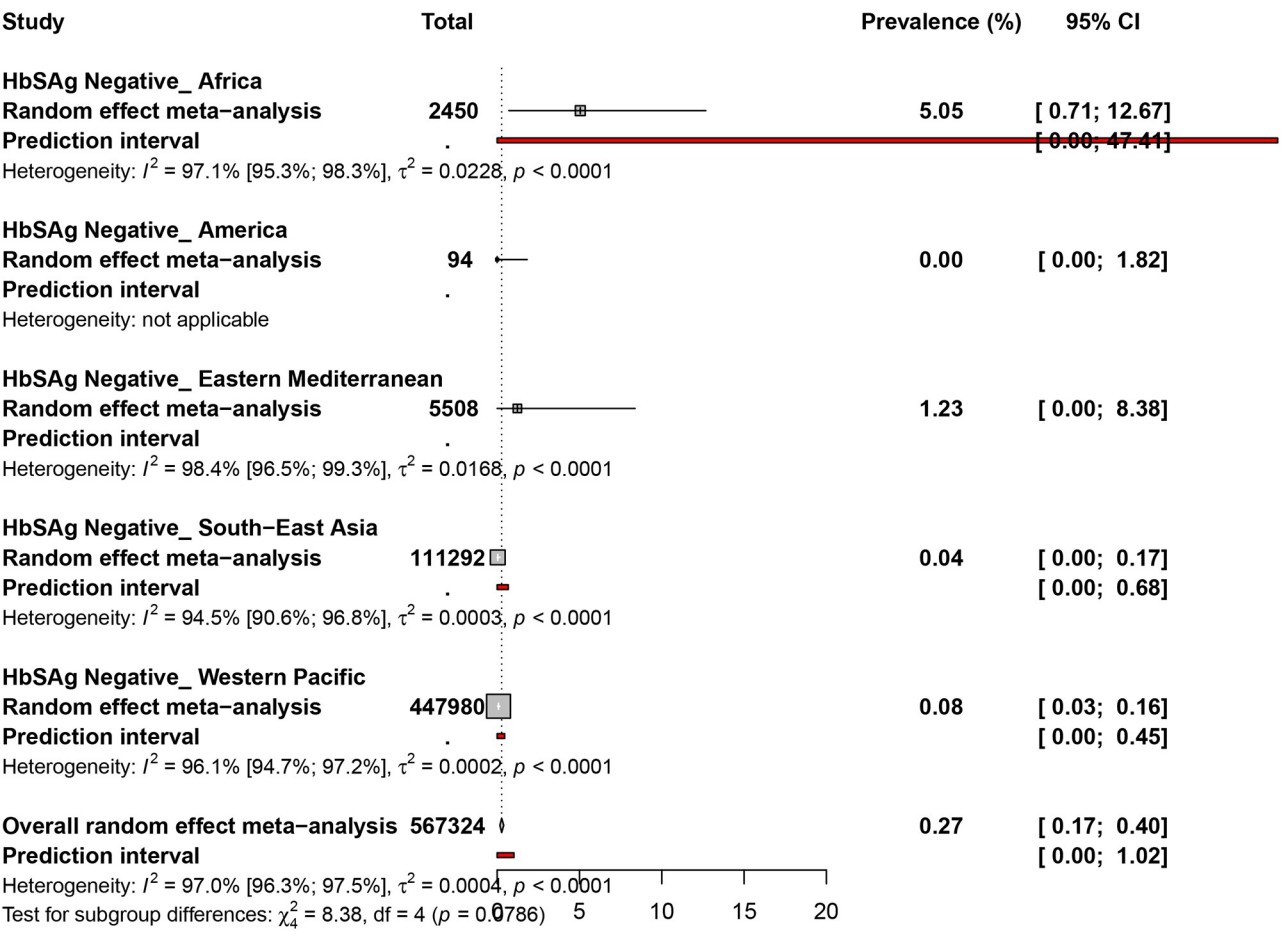

**Fig 4. The pooled global prevalence of hepatitis B virus infection in HBsAg negative blood donors.**

OBI in blood donors [31, 32]. The first study reported a prevalence of 16.4% in 4 studies of blood donors in Sudan [31]. The second study through a Bayesian meta-analysis determined a prevalence of 9.4% among blood donors in China [31]. Of course, our global study cannot be compared to these national systematic reviews. In addition, these two previous systematic reviews did not differentiate the subject with OBI from those in the negative window period [33]. The pooled prevalence of OBI in blood donors was highest in the regions of Eastern Mediterranean, South-East Asia, and Africa. These regions are mainly represented by developing countries which are also areas with high endemicity of HBV. These results are similar to Liu et al., study who reported a higher prevalence of OBI in low-income regions of China [32]. The prevalence of OBI shows great variability in different regions of the world depending on the study population, study design, HBV endemicity, and several factors [12, 34, 35]. Among these factors are the diagnostic approaches used for the detection of anti-HBc, anti-HBs, HBsAg and DNA which was not fully explored in this study. Although studies which explicitly stated that some participants had repeat donations were excluded, this phenomenon is common in most blood banks and therefore cannot guarantee that a residue of this limit did not influence the estimates in this review. Replacement and paid donors are also factors that affect the possibility of having repeated donors in the estimates. Although our review was limited to English or French, this study is the first to report the prevalence of OBI and the negative window period in blood donors with representativeness from all WHO regions of the world.

**Table 1. Summary of meta-analysis results for the prevalence of occult hepatitis B in blood donors.**

| | Prevalence. % (95%CI) | 95% Prediction interval | N Studies | N Participants | ¶H (95% CI) | §I² (95%CI) | P heterogeneity |
|---|---|---|---|---|---|---|---|
| **Occult hepatitis B virus prevalence in HBsAg negative & anti-HBc positive blood donors** | | | | | | | |
| Overall | 6.6 [5.7–7.5] | [2.2–12.8] | 53 | 6756446 | 10.9 [10.5–11.5] | 99.2 [99.1–99.2] | <0.001 |
| Cross-sectional | 6.6 [5.7–7.5] | [2.2–12.8] | 53 | 6756446 | 10.9 [10.5–11.5] | 99.2 [99.1–99.2] | <0.001 |
| Low risk of bias | 9.9 [7.7–12.3] | [1.1–25.2] | 30 | 6503856 | 12.6 [11.9–13.3] | 99.4 [99.3–99.4] | <0.001 |
| **Occult hepatitis B virus prevalence in HBsAg negative & anti-HBc negative blood donors** | | | | | | | |
| Overall | 0 [0–0] | [0–0.2] | 7 | 72555 | 1.4 [1–2.1] | 46.7 [0–77.5] | 0.081 |
| Cross-sectional | 0 [0–0] | [0–0.2] | 7 | 72555 | 1.4 [1–2.1] | 46.7 [0–77.5] | 0.081 |
| Low risk of bias | 0 [0–0] | [0–0] | 4 | 72107 | 1 [1–2.6] | 0 [0–84.7] | 0.731 |
| **Occult hepatitis B virus prevalence in HBsAg negative blood donors** | | | | | | | |
| Overall | 0.3 [0.2–0.4] | [0–1] | 27 | 567324 | 5.8 [5.2–6.4] | 97 [96.3–97.5] | <0.001 |
| Cross-sectional | 0.2 [0.1–0.3] | [0–0.9] | 26 | 566816 | 5.7 [5.1–6.3] | 96.9 [96.2–97.5] | <0.001 |
| Low risk of bias | 0.4 [0.2–0.5] | [0–1.2] | 15 | 467345 | 6.7 [5.9–7.5] | 97.7 [97.1–98.2] | <0.001 |

CI: confidence interval; N: Number; 95% CI: 95% Confidence Interval; NA: not applicable.

¶H is a measure of the extent of heterogeneity, a value of H = 1 indicates homogeneity of effects and a value of H >1indicates a potential heterogeneity of effects.

§: I2 describes the proportion of total variation in study estimates that is due to heterogeneity, a value > 50% indicates presence of heterogeneity

Molecular techniques made a significant contribution in reducing the risk of transmission of HBV through blood transfusion. The continuous improvement in the sensitivity of molecular tests further reduced the seronegative window period. The high cost, facilities, qualified personnel represent the major obstacles which slow down the adoption of molecular techniques in low-resource settings which paradoxically show the highest rates of HBV and OBI infections [36]. The introduction of molecular techniques into pretransfusion algorithms across all regions of the world and more particularly low-resource areas should be initiated to ensure safer transfusion [37]. The major considerations for the introduction of molecular testing in low-income regions include the consideration of additional number of units of discarded blood with its impact on the shortage of blood units. Also, the development of more sensitive and specific molecular tests is necessary for continuous improvement. In this sense, studies that aim to evaluate the newly developed tests should be implemented. Moreso, it would be important to adopt the vaccination of HBV negative donors without anti-HBs and apply vaccine policies that consider all population categories, is order to achieve the goal of eradicating HBV infection by 2030.

## 5. Conclusion

This review shows a high prevalence of OBI and / or HBV in the negative window period with the highest frequencies recorded in low-resource areas.

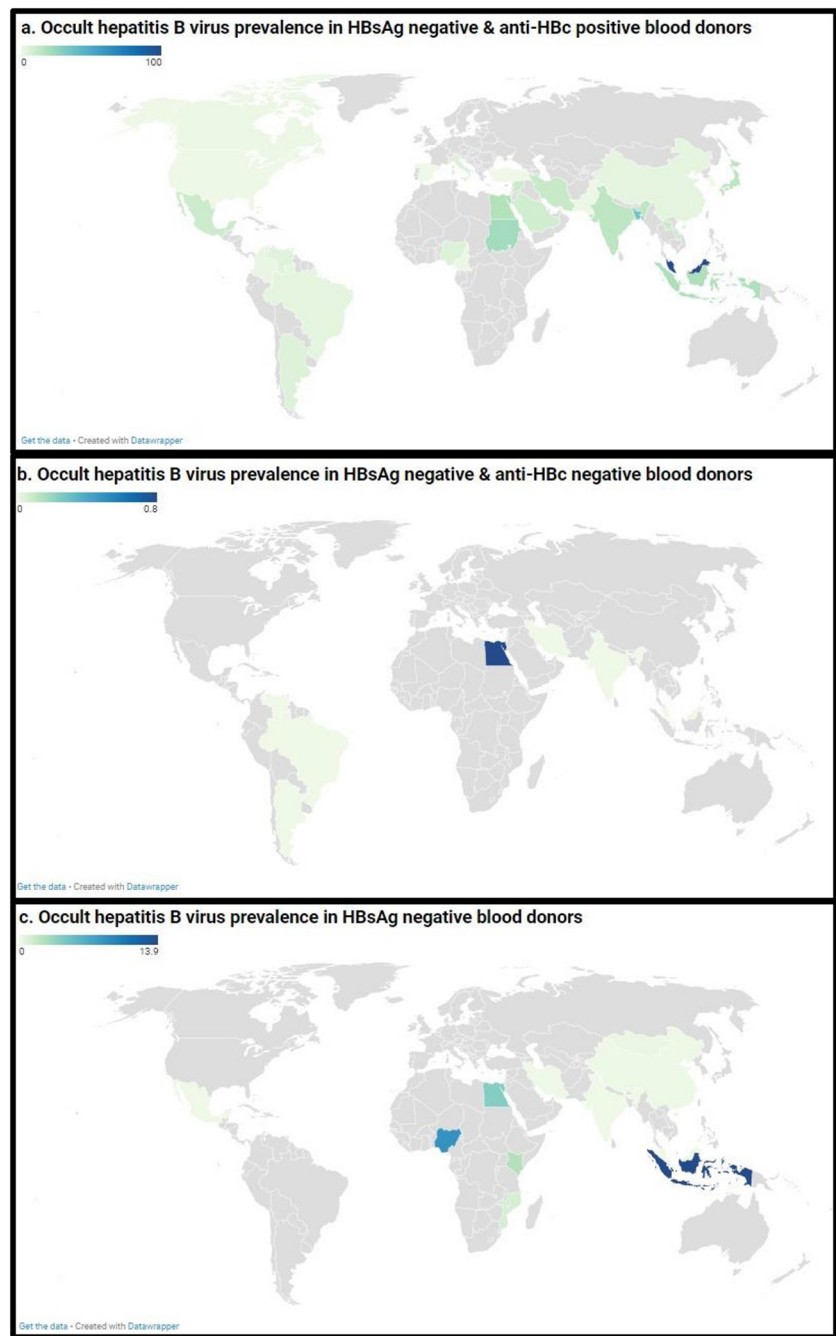

**Fig 5. Global prevalence estimate of occult hepatitis B virus in blood donors.** The letters (a, b, and c) denote blood donors with HBsAg negative and anti-HBc positive, HBsAg negative and anti-HBc negative, and HBsAg negative, respectively. Map source: https://www.datawrapper.de/.

## Supporting information

**S1 Appendix. Preferred reporting items for systematic reviews and meta-analyses checklist.**
(PDF)

**S2 Appendix. Search strategy in PubMed.**
(PDF)

**S3 Appendix. Items for risk of bias assessment.**
(PDF)

**S4 Appendix. Main reasons of exclusion of eligible studies.**
(PDF)

**S5 Appendix. Reference list of included studies on global prevalence of occult hepatitis B infection in blood donors.**
(PDF)

**S6 Appendix. Characteristics of included studies.**
(PDF)

**S7 Appendix. Individual characteristics of included studies.**
(PDF)

**S8 Appendix. Risk of bias assessment.**
(PDF)

**S9 Appendix. The pooled global prevalence of hepatitis B virus infection in HBsAg negative and anti-HBc positive blood donors.**
(PDF)

**S10 Appendix. The pooled global prevalence of hepatitis B virus infection in HBsAg negative blood donors.**
(PDF)

**S11 Appendix. Funnel chart for publications of the occult hepatitis B virus prevalence in HBsAg negative and anti-HBc positive blood donors.**
(PDF)

**S12 Appendix. Funnel chart for publications of the occult hepatitis B virus prevalence in HBsAg negative and anti-HBc negative blood donors.**
(PDF)

**S13 Appendix. Funnel chart for publications of the occult hepatitis B virus prevalence in HBsAg negative blood donors.**
(PDF)

**S14 Appendix. Subgroup analyses of global prevalence of occult hepatitis B virus in blood donors.**
(PDF)

**S1 Checklist. Preferred reporting items for systematic reviews and meta-analyses checklist.**
(PDF)

## Author Contributions

**Conceptualization:** Guy Roussel Takuissu, Sebastien Kenmoe, Lucy Ndip.

**Data curation:** Guy Roussel Takuissu, Sebastien Kenmoe, Marie Amougou Atsama, Donatien Serge Mbaga, Jean Thierry Ebogo-Belobo, Arnol Bowo-Ngandji, Martin Gael Oyono, Jeannette Nina Magoudjou-Pekam, Ginette Irma Kame-Ngasse, Elisabeth Zeuko'o

Menkem, Abdel Aziz Selly Ngaloumo, Agnès Thierry Rebecca Banlock, Alfloditte Flore Feudjio, Cromwel Zemnou-Tepap, Dowbiss Meta-Djomsi, Gilberte Louise Nyimbe Mviena, Ines Nyebe Eloundou, Jacqueline Félicité Yéngué, Josiane Kenfack-Zanguim, Juliette Laure Ndzie Ondigui, Ridole Martin Zekeng Mekontchou, Sabine Aimee Touangnou-Chamda, Yrene Kamtchueng Takeu, Jean Bosco Taya-Fokou, Chris Andre Mbongue Mikangue, Raoul Kenfack-Momo, Cyprien Kengne-Nde.

**Formal analysis:** Sebastien Kenmoe, Cyprien Kengne-Nde.

**Funding acquisition:** Sebastien Kenmoe.

**Methodology:** Guy Roussel Takuissu, Sebastien Kenmoe, Marie Amougou Atsama, Etienne Atenguena Okobalemba, Donatien Serge Mbaga, Jean Thierry Ebogo-Belobo, Arnol Bowo-Ngandji, Martin Gael Oyono, Jeannette Nina Magoudjou-Pekam, Ginette Irma Kame-Ngasse, Elisabeth Zeuko'o Menkem, Abdel Aziz Selly Ngaloumo, Agnès Thierry Rebecca Banlock, Alfloditte Flore Feudjio, Cromwel Zemnou-Tepap, Dowbiss Meta-Djomsi, Gilberte Louise Nyimbe Mviena, Ines Nyebe Eloundou, Jacqueline Félicité Yéngué, Josiane Kenfack-Zanguim, Juliette Laure Ndzie Ondigui, Ridole Martin Zekeng Mekontchou, Sabine Aimee Touangnou-Chamda, Yrene Kamtchueng Takeu, Jean Bosco Taya-Fokou, Chris Andre Mbongue Mikangue, Raoul Kenfack-Momo, Cyprien Kengne-Nde, Seraphine Nkie Esemu, Richard Njouom.

**Project administration:** Sebastien Kenmoe, Lucy Ndip.

**Supervision:** Sebastien Kenmoe, Lucy Ndip.

**Validation:** Guy Roussel Takuissu, Sebastien Kenmoe, Marie Amougou Atsama, Etienne Atenguena Okobalemba, Donatien Serge Mbaga, Jean Thierry Ebogo-Belobo, Arnol Bowo-Ngandji, Martin Gael Oyono, Jeannette Nina Magoudjou-Pekam, Ginette Irma Kame-Ngasse, Elisabeth Zeuko'o Menkem, Abdel Aziz Selly Ngaloumo, Agnès Thierry Rebecca Banlock, Alfloditte Flore Feudjio, Cromwel Zemnou-Tepap, Dowbiss Meta-Djomsi, Gilberte Louise Nyimbe Mviena, Ines Nyebe Eloundou, Jacqueline Félicité Yéngué, Josiane Kenfack-Zanguim, Juliette Laure Ndzie Ondigui, Ridole Martin Zekeng Mekontchou, Sabine Aimee Touangnou-Chamda, Yrene Kamtchueng Takeu, Jean Bosco Taya-Fokou, Chris Andre Mbongue Mikangue, Raoul Kenfack-Momo, Cyprien Kengne-Nde, Seraphine Nkie Esemu, Richard Njouom, Lucy Ndip.

**Writing – original draft:** Guy Roussel Takuissu, Sebastien Kenmoe.

**Writing – review & editing:** Guy Roussel Takuissu, Sebastien Kenmoe, Marie Amougou Atsama, Etienne Atenguena Okobalemba, Donatien Serge Mbaga, Jean Thierry Ebogo-Belobo, Arnol Bowo-Ngandji, Martin Gael Oyono, Jeannette Nina Magoudjou-Pekam, Ginette Irma Kame-Ngasse, Elisabeth Zeuko'o Menkem, Abdel Aziz Selly Ngaloumo, Agnès Thierry Rebecca Banlock, Alfloditte Flore Feudjio, Cromwel Zemnou-Tepap, Dowbiss Meta-Djomsi, Gilberte Louise Nyimbe Mviena, Ines Nyebe Eloundou, Jacqueline Félicité Yéngué, Josiane Kenfack-Zanguim, Juliette Laure Ndzie Ondigui, Ridole Martin Zekeng Mekontchou, Sabine Aimee Touangnou-Chamda, Yrene Kamtchueng Takeu, Jean Bosco Taya-Fokou, Chris Andre Mbongue Mikangue, Raoul Kenfack-Momo, Cyprien Kengne-Nde, Seraphine Nkie Esemu, Richard Njouom, Lucy Ndip.

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
