## [Decision Letter · Decision Letter 0]

16 Mar 2022

PONE-D-22-03546Global epidemiology of occult hepatitis B virus infections in blood donors, a systematic review and meta-analysisPLOS ONE

Dear Dr. Kenmoe,

Thank you for submitting your manuscript to PLOS ONE. After careful consideration, we feel that it has merit but does not fully meet PLOS ONE’s publication criteria as it currently stands. Therefore, we invite you to submit a revised version of the manuscript that addresses the points raised during the review process.

We look forward to receiving your revised manuscript.

Kind regards,

Jason T. Blackard, PhD

Academic Editor

PLOS ONE

“This project is part of the EDCTP2 programme supported by the European Union.”

“This project is part of the EDCTP2 programme supported by the European Union under grant agreement TMA2019PF-2705. The funders had no role in study design, data collection and analysis, decision to publish, or preparation of the manuscript.”

3. We note that [Figure 4] in your submission contain [map/satellite] images which may be copyrighted. All PLOS content is published under the Creative Commons Attribution License (CC BY 4.0), which means that the manuscript, images, and Supporting Information files will be freely available online, and any third party is permitted to access, download, copy, distribute, and use these materials in any way, even commercially, with proper attribution. For these reasons, we cannot publish previously copyrighted maps or satellite images created using proprietary data, such as Google software (Google Maps, Street View, and Earth). For more information, see our copyright guidelines: http://journals.plos.org/plosone/s/licenses-and-copyright.

a. You may seek permission from the original copyright holder of [Figure 4] to publish the content specifically under the CC BY 4.0 license. 

Natural Earth (public domain): http://www.naturalearthdata.com/.

Additional Editor Comments:

This is a meta-analysis of occult HBV in blood donors.

There are multiple awkward phrases, and the manuscript should be revised carefully by a native English speaker and/or a professional editing service.

The introduction includes several unsubstantiated sentences and requires additional references and/or restructuring.  For example:

·      These two factors contribute to making HBV the virus whose transmission during blood transfusion is the most frequent [6]. . . . confusing wording / should be reworded

·      Occult HBV infections are explained by several mechanisms including reduction . . . what does “reduction” mean here?

·      In high-income countries, weakly endemic for HBV,  . . . “weakly endemic” is not the correct term

·      Screening for anti-HBc is generally associated with insufficient blood in areas highly endemic for HBV . . . does “insufficient blood” mean insufficient volume of blood for testing or something else?

·      Although OBI is not always transmissible through blood transfusion, several studies have reported cases of post-transfusion infections with OBI and during the seronegative window period [5, 21-23]. . . . there are plenty of studies suggesting that OBI can be transferred to another individual who then develops OBI or chronic HBV, so then what is the meaning of this sentence?

·      The risk of transmission is also very high in immunocompromised recipients or multiple transfused subjects [1]. . . . very high compared to what?

The authors are confusing the seronegative window period with occult HBV infection.  These are not the same thing and should not be combined into one analysis.  This is a major limitation and must be addressed prior to publication.

What does UNSD region stand for?  The full name is never provided.

What does “low risk of bias” mean for some studies?

In figure 1, the N for the “additional records identified through other sources” is absent.

In figure 2, change the prevalence scale from 0 to 100 to 0 to ~20 would be more helpful to visualize the results.

Appendix 4 should be updated with consistent formatting – capitalization of the author’s name, no use of all caps, etc.

Reviewers' comments:

Reviewer's Responses to Questions

**Comments to the Author**

1. Is the manuscript technically sound, and do the data support the conclusions?

Reviewer #1: Yes

Reviewer #2: Partly

2. Has the statistical analysis been performed appropriately and rigorously? 

Reviewer #1: Yes

Reviewer #2: Yes

3. Have the authors made all data underlying the findings in their manuscript fully available?

Reviewer #1: Yes

Reviewer #2: Yes

4. Is the manuscript presented in an intelligible fashion and written in standard English?

Reviewer #1: Yes

Reviewer #2: No

5. Review Comments to the Author

Reviewer #1: This is a well-conducted analysis that would be a useful addition to the literature. I suggest the authors add further context for the observed heterogeneity. While the I2 values are very large (meaning that the vast majority of variation is due to differences among studies, rather than imprecision within individual studies), this is not surprising because many of the studies are quite large. As the sample sizes become large within-study variation shrinks toward zero, so I2 must increase even if the absolute degree of among-study variation is negligible. In some cases the individual estimates appear similar but I2 is very large. The authors should comment on whether the large "relative" variation is accompanied by enough "absolute" variation to be meaningful (worth attempting to explain by examining factors that differed among studies).

Reviewer #2: This manuscript began with the laudable objective of determined OBI prevalence worldwide among blood donors. It may be a question of semantics, but as the article search was limited to English or French languages, it is difficult to assume that global prevalence of occult HBV could be accurately estimated. There were multiple other weaknesses that dampened my enthusiasm. For example, testing of donor pool blood samples for HBV DNA is the most critical piece of OBI detection; yet the discussion states that this was not a point of examination for the current manuscript. I think it is difficult to draw meaningful conclusions without a good assessment of the diagnostic methods used for HBV DNA at the least (not to mention the serologies). Moreover, I had a great deal of difficulty reading the article. The term “prevalence” was used to describe numbers of articles from different countries; that is an incorrect use of the term, and was quite confusing. Similarly, there are contradictory statements made: e.g., most of the data came from cross-sectional studies, and most of the data came from prospective sampling (which would not constitute a cross-sectional study). I did not see an assessment of study quality, which is a key factor when conducting a meta-analysis. The fact that many studies did not report age certainly renders quality suspect. Net: it is not clear that the conclusions in this manuscript are supported by the data.

6. PLOS authors have the option to publish the peer review history of their article (what does this mean?). If published, this will include your full peer review and any attached files.

Reviewer #1: No

Reviewer #2: No

---

## [Author Response · Author response to Decision Letter 0]

14 Apr 2022

Review Comments to the Author

“This project is part of the EDCTP2 programme supported by the European Union under grant agreementTMA2019PF-2705. The funders had no role in study design, data collection and analysis, decision to publish, or preparation of the manuscript.”

Authors: corrected, thank you. Our funding statement is as above. “This project is part of the EDCTP2 programme supported by the European Union under grant agreementTMA2019PF-2705. The funders had no role in study design, data collection and analysis, decision to publish, or preparation of the manuscript.”

We note that [Figure 4] in your submission contain [map/satellite] images which may be copyrighted. All PLOS content is published under the Creative Commons Attribution License (CC BY 4.0), which means that the manuscript, images, and Supporting Information files will be freely available online, and any third party is permitted to access, download, copy, distribute, and use these materials in any way, even commercially, with proper attribution. For these reasons, we cannot publish previously copyrighted maps or satellite images created using proprietary data, such as Google software (Google Maps, Street View, and Earth). For more information, see our copyright guidelines: http://journals.plos.org/plosone/s/licenses-and-copyright.

a. You may seek permission from the original copyright holder of [Figure 4] to publish the content specifically under the CC BY 4.0 license.

We recommend that you contact the original copyright holder with the Content Permission Form(http://journals.plos.org/plosone/s/file?id=7c09/content-permission-form.pdf) and the

following text:

b. If you are unable to obtain permission from the original copyright holder to publish these figures under the CC BY4.0 license or

if the copyright holder’s requirements are incompatible with the CC BY 4.0 license, please either i) remove the figure or ii) supply a replacement figure that complies with the CC BY 4.0 license. Please check copyright information on all replacement figures and update the figure caption with source information. If applicable, please specify in the figure caption text when a figure is similar but not identical to the original image and is therefore for illustrative purposes only.

Maps at the CIA (public domain): https://www.cia.gov/library/publications/the-world-factbook/index.html andhttps://www.cia.gov/library/publications/cia-maps-publications/index.html

NASA Earth Observatory

(public domain): http://earthobservatory.nasa.gov/

Natural Earth (public domain): http://www.naturalearthdata.com/.

Authors: We checked the license terms of Datawrapper (https://www.datawrapper.de/: the site we created the figure with), and it looks like they request the label "Created with Datawrapper" that you find below our Fig. 4 to be kept on the figure. We therefore assure you that we meet the CC BY 4.0 license requirements for our fig. 4 for this article.

This is a meta-analysis of occult HBV in blood donors. There are multiple awkward phrases, and the manuscript should be revised carefully by a native English speaker and/or a professional editing service.

The introduction includes several unsubstantiated sentences and requires additional references and/or restructuring. For example:

· These two factors contribute to making HBV the virus whose transmission during blood transfusion is the most frequent [6]. . . . confusing wording / should be reworded

· Occult HBV infections are explained by several mechanisms including reduction . . . what does “reduction” mean here?

· In high-income countries, weakly endemic for HBV, . . . “weakly endemic” is not the correct term

· Screening for anti-HBc is generally associated with insufficient blood in areas highly endemic for HBV . . . does “insufficient blood” mean insufficient volume of blood for testing or something else?

· Although OBI is not always transmissible through blood transfusion, several studies have reported cases of post-transfusion infections with OBI and during the seronegative window period [5, 21-23]. . . . there are plenty of studies suggesting that OBI can be transferred to another individual who then develops OBI or

chronic HBV, so then what is the meaning of this sentence?

· The risk of transmission is also very high in immunocompromised recipients or multiple transfused subjects [1]. .. . very high compared to what?

Authors: We thank the reviewer for all the comments. The manuscript has been improved

according to the comments and suggestions.

The authors are confusing the seronegative window period with occult HBV infection. These are not the same thing and should not be combined into one analysis. This is a major limitation and must be addressed prior to publication.

Authors: Thank you for the comment. We reread all included articles and extracted data when possible, distinguishing studies with positive (occult hepatitis B virus) and negative (window period) anti-HBc. We also repeated all the analyses.

What does UNSD region stand for? The full name is never provided.

Authors: Thank you corrected, United Nations Statistics Division.

What does “low risk of bias” mean for some studies?

Authors: We used the scale proposed by Hoy et al (doi: 10.1016/j.jclinepi.2011.11.014) to assess the risk of bias in prevalence studies. Each study is evaluated on 10 questions (10 marks) which makes it possible to categorize all the included studies in low, moderate or high risk of bias. We also conducted a sensitivity analysis including only studies with a low risk of bias.

In figure 1, the N for the “additional records identified through other sources” is absent.

Authors: corrected as suggested, thank you.

In figure 2, change the prevalence scale from 0 to 100 to 0 to ~20 would be more helpful to visualize the results.

Authors: Fig. 2 is the simplified version of the Appendix S9. We have provided this simplified version since the full version is larger than the A4 printable format. The figure actually hides multiple underlying information such as the prevalence of individual studies which are above 20 and which prevent us from being able to reduce the scale. Thanks for the suggestion. 

Appendix 4 should be updated with consistent formatting – capitalization of the author’s name, no use of all caps, etc.

Authors: corrected as suggested, thank you.

Reviewer #1: This is a well-conducted analysis that would be a useful addition to the literature. 

Authors: Thank you for this appreciation.

I suggest the authors add further context for the observed heterogeneity. While the I2 values are very large (meaning that the vast majority of variation is due to differences among studies, rather than imprecision within individual studies), this is not surprising because many of the studies are quite large. As the sample sizes become large within-study variation shrinks toward zero, so I2 must increase even if the absolute degree of among-study variation is negligible. In some cases, the individual estimates appear similar but I2 is very large. The authors should comment on whether the large "relative" variation is accompanied by enough "absolute" variation to be meaningful (worth attempting to explain by examining factors that differed among studies).

Authors: We thank the Reviewer for this relevant comment. Heterogeneity is known to be inherent in meta-analyses of observational studies. We also thank the reviewer for pointing out that most of the variability is due to differences between studies and not differences within individual studies. To reduce the influence of sources of heterogeneity on our estimates, we distinguished, when possible, between negative and positive anti-HBc donors. In discussion section, we strengthened the explanation of potential sources of heterogeneity previously mentioned such as the population studied with parameters such as age, type of donors (voluntary, replacement, remunerated, etc.), the possibility of having repeat donors, endemicity of HBV, study design and diagnostic approach to HBV markers (anti-HBc, anti-HBs, HBsAg, and DNA).

Reviewer #2: This manuscript began with the laudable objective of determined OBI prevalence worldwide among blood donors. It may be a question of semantics, but as the article search was limited to English or French languages, it is difficult to assume that global prevalence of occult HBV could be accurately estimated. 

Authors: Our review is influenced by multiple factors including the language restriction which we reported in the discussion section, thank you.

There were multiple other weaknesses that dampened my enthusiasm. For example, testing of donor pool blood samples for HBV DNA is the most critical piece of OBI detection; yet the discussion states that this was not a point of examination for the current manuscript. 

Authors: For studies that tested pooled blood samples, we included only those that resolved positive pools by further analysis of individual samples as reported in the methodology, thank you.

I think it is difficult to draw meaningful conclusions without a good assessment of the diagnostic methods used for HBV DNA at the least (not to mention the serologies). 

Authors: The OBI considers only HBsAg negative subjects with positive and/or negative results for anti-HBc and anti-HBs. We agree on the great variability of sensitivity or specificity of HBV serological tests. This suggests the possibility of having HBsAg false-negative subjects recruited in some included studies. We mentioned these aspects as the limitations of the study which are responsible for the observed heterogeneity. We conducted a subgroup analysis according to the DNA detection test which unfortunately cannot allow us to draw a conclusion although the difference according to the techniques was statistically significant. The assay with the highest prevalence, LAMP, being represented by a single prevalence data.

Moreover, I had a great deal of difficulty reading the article. The term “prevalence” was used to describe numbers of articles from different countries; that is an incorrect use of the term, and was quite confusing. Similarly, there are contradictory statements made: e.g., most of the data came from cross-sectional studies, and most of the data came from prospective sampling (which would not constitute a cross-sectional study). 

Authors: A study could present several prevalence data according to the number of population categories. We clarified this aspect in the methodology, thank you.

I did not see an assessment of study quality, which is a key factor when conducting a meta-analysis. 

Authors: We presented the results of the risk of bias assessment at the end of the study characteristics (results section) and in Appendix S8

The fact that many studies did not report age certainly renders quality suspect. Net: it is not clear that the conclusions in this manuscript are supported by the data.

Authors: Although very few of the included studies reported donor age information, most blood transfusion policy worldwide relies on adult donors. We have now mentioned that this factor could be a potential source of heterogeneity in our estimates, thank you.

---

## [Decision Letter · Decision Letter 1]

29 Jul 2022

Global epidemiology of occult hepatitis B virus infections in blood donors, a systematic review and meta-analysis

PONE-D-22-03546R1

Dear Dr. Kenmoe,

We’re pleased to inform you that your manuscript has been judged scientifically suitable for publication and will be formally accepted for publication once it meets all outstanding technical requirements.

Kind regards,

Jason T. Blackard, PhD

Academic Editor

PLOS ONE

Additional Editor Comments (optional):

None

Reviewers' comments:

Reviewer's Responses to Questions

**Comments to the Author**

1. If the authors have adequately addressed your comments raised in a previous round of review and you feel that this manuscript is now acceptable for publication, you may indicate that here to bypass the “Comments to the Author” section, enter your conflict of interest statement in the “Confidential to Editor” section, and submit your "Accept" recommendation.

Reviewer #1: All comments have been addressed

2. Is the manuscript technically sound, and do the data support the conclusions?

Reviewer #1: (No Response)

3. Has the statistical analysis been performed appropriately and rigorously? 

Reviewer #1: (No Response)

4. Have the authors made all data underlying the findings in their manuscript fully available?

Reviewer #1: (No Response)

5. Is the manuscript presented in an intelligible fashion and written in standard English?

Reviewer #1: (No Response)

6. Review Comments to the Author

Reviewer #1: (No Response)

7. PLOS authors have the option to publish the peer review history of their article (what does this mean?). If published, this will include your full peer review and any attached files.

Reviewer #1: No

---

## [Editor Report · Acceptance letter]

12 Aug 2022

PONE-D-22-03546R1 

Global epidemiology of occult hepatitis B virus infections in blood donors, a systematic review and meta-analysis 

Dear Dr. Kenmoe:

I'm pleased to inform you that your manuscript has been deemed suitable for publication in PLOS ONE. Congratulations! Your manuscript is now with our production department. 

Kind regards, 

on behalf of

Dr. Jason T. Blackard 

Academic Editor

PLOS ONE